# Comparability of modern contraceptive use estimates between a face-to-face survey and a cellphone survey among women in Burkina Faso

**Abigail R. Greenleaf** [1]*, **Aliou Gadiaga**[2‡], **Georges Guiella**[2‡], **Shani Turke**[1‡], **Noelle Battle**[1‡], **Saifuddin Ahmed**[1], **Caroline Moreau**[1,3]

1 Department of Population Family and Reproductive Health, Johns Hopkins University, Baltimore, MD, United States of America, 2 Institut Supérieur des Sciences de la Population, University of Ouagadougou, Institut Supérieur des Sciences de la Population, Ouagadougou, Burkina Faso, 3 Gender, Sexual and Reproductive Health, Centre for Research in Epidemiology and Population Health (CESP), Villejuif, France

☯ These authors contributed equally to this work.
‡ These authors also contributed equally to this work.
* agreenleaf@jhu.edu

**Data Availability Statement:** Data are third party data owned by PMA2020. Data are available upon request from PMA2020's website: https://www.

## Abstract

### Introduction

The proliferation of cell phone ownership in Sub-Saharan Africa (SSA) presents the opportunity to collect public health indicators at a lower cost compared to face-to-face (FTF) surveys. This analysis assesses the equivalence of modern contraceptive prevalence estimates between a nationally representative FTF survey and a cell phone survey using random digit dialing (RDD) among women of reproductive age in Burkina Faso.

### Methods

We analyzed data from two surveys conducted in Burkina Faso between December 2017 and May 2018. The FTF survey conducted by Performance Monitoring and Accountability (PMA2020) comprised a nationally representative sample of 3,556 women of reproductive age (15–49 years). The RDD survey was conducted using computer-assisted telephone interviewing and included 2,379 women of reproductive age.

### Results

Compared to FTF respondents, women in the RDD sample were younger, were more likely to have a secondary degree and to speak French. RDD respondents were more likely to report using modern contraceptive use (40%) compared to FTF respondents (26%) and the difference remained unchanged after applying post-stratification weights to the RDD sample (39%). This difference surpassed the equivalence margin of 4%. The RDD sample also produced higher estimates of contraceptive use than the subsample of women who owned a phone in the FTF sample (32%). After adjusting for women's sociodemographic factors, the odds of contraceptive use were 1.9 times higher (95% CI: 1.6–2.2) in the RDD survey

pma2020.org/request-access-to-datasets. The authors did not receive special access privileges to the data that others would not have.

**Funding:** This work was supported by the Bill & Melinda Gates Foundation (www.gatesfoundation. org). All authors were funded through award # OPP1079004. The findings and conclusions are those of the authors and do not necessarily reflect positions or policies of the Bill & Melinda Gates Foundation. The funders had no role in study design, data collection and analysis, decision to publish, or preparation of the manuscript.

**Competing interests:** The authors have declared that no competing interests exist.

compared to the FTF survey and 1.6 times higher (95% CI: 1.3–1.8) compared to FTF phone owners.

## Conclusions

Modern contraceptive prevalence in Burkina Faso is over-estimated when using a cell phone RDD survey, even after adjusting for a number of sociodemographic factors. Further research should explore causes of differential estimates of modern contraceptive use by survey modes.

## Introduction

Largely due to inadequate vital registration systems, data from population-based surveys such as the Demographic and Health Surveys (DHS) and UNICEF's Multiple Indicator Cluster Survey (MICS) serve as the primary data source in low and middle-income countries to monitor progress towards the Sustainable Development Goals and track changes in population health [1].

Although the DHS and MICS have provided invaluable data for the past 30 years, there are two main challenges associated with face-to-face (FTF) surveys: cost and extensive field time to attain the needed population coverage [2, 3]. The challenges of FTF surveys in low- and middle- income countries (LMIC) are contrasted with the three primary advantages of phone surveys identified in the 1990s that remain pertinent today: speed of data collection, cost efficiency, and ability to supervise interviewers throughout data collection [4]. Cost reduction is a major consideration in LMIC; one study conducted in Honduras showed a decrease from $40 USD per interview using FTF to $17 USD using a cellphone survey with interactive voice response (IVR) [5].

Cell phone ownership grew exponentially in Sub-Saharan Africa (SSA) in the early 2000s, driven mainly by urbanization and low cost of purchasing a cell phone and airtime [6, 7]. There are currently over 444 million cell phone subscriptions in SSA and over 634 million subscriptions are anticipated by 2025 [6]. SSA is expected to reach 50% mobile phone penetration, or the percent of unique mobile phone users within a specific population, in 2021 and 52% penetration by 2025 [6, 8]. In this context, demographers are considering the use of cell phone surveys to track national health indicators [9–11]. However, phone survey estimates need to be compared with FTF results before substituting data collection modes, to ensure phone survey estimates are valid. Threats to survey data quality and validity can be categorized into five types of survey error: frame and non-response errors, which relate to survey representation, and specification, measurement and processing errors, which relate to the quality of data collected. The manifestation of these errors differs by data collection mode and sampling approach [12]. Cell phone surveys are traditionally collected using one of three modes–computer assisted telephone interviews (CATI), IVR or short message service (SMS). The focus of this study is survey errors when comparing CATI to FTF surveys.

Few studies comparing cell phone surveys to FTF studies have been conducted in LMIC [13, 14]. Those that have compared cell phone and FTF survey results used a census or a FTF reference population-based survey such as DHS, rather than a concurrent study with the same questions in the cell phone survey. Two recent random digit dial (RDD) studies, an IVR survey in Ghana about general health [10] and a CATI survey in Cote d'Ivoire about HIV risk behaviors [15] compared their population with the sociodemographic composition of recent

national FTF surveys. In Ghana, two-thirds of RDD respondents were male and more than half were 15–24 years of age whereas according to the 2017 census, only 48% of the population is male and 30% ages 15–24 [10]. In the Cote d'Ivoire study, the composition of the RDD sample was different than the DHS sample distribution, with over-representation of urban individuals and males [15].

The aim of this analysis is to examine whether an RDD cell phone survey produces equivalent estimates of modern contraceptive use to a reference FTF survey conducted in the general female population after applying post-stratification weights to account for RDD sample distortion.

## Materials and methods

### Methods

The PMA2020 FTF survey has exempt status (IRB #00000287, exempt as "public health practice", defined by DHHS regulations 45 CFR 46.102), as determined by John Hopkins' IRB. Approval was granted in July 2014 by the Comité d'éthique pour la recherche en santé (IRB #2014-7-81). Consent was obtained orally. For the phone survey, ethical approval was granted by the IRB at the Johns Hopkins Bloomberg School of Public Health (IRB No. 00007961). The ISSP team submitted a protocol to their ethical committee for the RDD survey (IRB No. 2018-3-036).

We used two datasets for this analysis. The first dataset is the Burkina Faso PMA2020 Round 5 (R5) FTF survey, designed to track key family planning indicators under the Family Planning 2020 initiative [16]. PMA2020 R5 implemented a two-stage stratified cluster design to select a national sample of women of reproductive age in Burkina Faso. This design starts with a probability proportional to size selection of 83 geographic clusters stratified by urban and rural areas followed by a random selection of about 35 households within each sample cluster. Detailed sampling methods and procedures are available in a previous publication [17]. PMA2020 R5 Burkina Faso was conducted between November 20, 2017- January 20, 2018 and included a total of 2,811 households (98.5% response rate) and 3,659 females (97.8% response rate) [18].

Data collection was performed by a network of trained female interviewers who conducted FTF interviews with members of selected households and with all eligible females 15–49 years from the selected households. Interviewers recorded responses directly on cell phones and uploaded the data when cellular network was available.

The second dataset was the RDD CATI survey, which took place four months after PMA2020 R5 survey. To conduct the RDD CATI survey, PMA2020 Baltimore and *Institute Supérieur des Sciences de la Population* (ISSP) staff trained five call center supervisors for three days. Supervisors had at least a Bachelor degree and all had previous survey research experience. Twenty-five interviewers were trained for four days and 20 interviewers were retained for data collection. All interviewers spoke French and at least one of four local languages, except for one interviewer who spoke exclusively French. Data collection was led by ISSP and took place from April 13 to May 17, 2018 in a call center located in an NGO building in Ouagadougou. Interviewers worked in two shifts, the first group from approximately 12–4 pm, the second group from 4–8 pm.

CATI was chosen over IVR based on a previous PMA2020 study which found higher response rates and less sample distortion in a CATI sample than a IVR sample [19]. RDD sampling was chosen because the mobile network operators in Burkina Faso do not share lists of valid phone numbers. To create the phone numbers, Viamo, an international mobile technology survey company, [20] used a list of the 25 existing prefixes provided from the three mobile

network operators in Burkina Faso. Viamo then randomly generated the remaining six digits to create a list of phone numbers.

Results from the aforementioned follow-up phone survey in Burkina Faso also informed the sampling strategy of the RDD cross-sectional survey, which included quotas by age and area of residence to improve representativeness and reduce design effects [21]. We established the quota groups by comparing respondents and non-respondents from a phone follow-up study of women who participated in the previous national FTF survey conducted by PMA2020 in 2017 (Round 4). Women who responded to the follow-up phone interview were more likely to live in an urban setting, had a higher level of education and were more likely to be over the age of 20. We defined quota groups based on residence and age for ease of implementation and used the distribution of these characteristics in the PMA2020 Round 4 female sample to calculate the percent of women in each of the four quota groups (see Table 1).

In the absence of active phone number lists, a significant proportion of phone numbers generated by RDD are invalid. A pilot study conducted in February 2018 estimated that 58% of RDD generated numbers were not assigned to a subscriber, or were invalid. We defined a phone number as valid if the outcome of the call, as recorded by any of the three mobile phone companies in Burkina Faso had one of three following outcomes (1- No Answer, 2- Normal Clearing or 3- Normal Unspecified) [22].

To improve RDD efficiency in light of the pilot results, we sent out an IVR "validation/pre-notification" phone call to all generated RDD numbers to eliminate invalid phone numbers. Calls were placed one to seven days before being called by an interviewer, with the following message in Moore: "Thank you for responding to our call. We will call you this week for a study. Please, pick up the phone when we call. Have a good day!"

For the first seven days of data collection, we called phone numbers that were identified with any one of the three call statuses during the validation calls: 1) No Answer, 2) Normal Clearing or 3) Normal Unspecified. However, after the first seven days of data collection, due to an insufficient number of completed interviews per day, we narrowed the definition of a valid phone number and no longer included phone numbers marked as 'No Answer' during the validation call. The narrower definition meant that interviewers were only calling phone numbers that were answered during the validation calls.

When a female respondent answered the call, she could either complete the survey at that time or be called back up to six times. If a respondent answered that did not speak the same language as the interviewer, she would be called back the same day by an interviewer that spoke the respondent's language. If a respondent explicitly refused the study, she was not called back. Men were not allowed to pass the phone to a female in their household. Women that completed the survey were sent the equivalent of $1 US dollar phone credit the day after completing the interview.

## Questionnaires

The PMA2020 Burkina Faso FTF female questionnaire includes standardized questions that are largely based on the DHS. The questionnaire typically takes less than 40 minutes to

**Table 1. The number of completed surveys needed by quota group.**

| Age groups | Rural | Urban |
|:---:|:---:|:---:|
| 15–19 | 388 (16.4%) | 130 (5.5%) |
| 20–49 | 1417 (60%) | 428 (18.1%) |
| Total | | 2363 |

complete [23]. The female survey collects sociodemographic information including cell phone ownership, and reproductive health measures, including current contraceptive use as described in the next section [17].

The RDD questionnaire was limited to 19 questions that allowed comparison of modern contraceptive use estimates with the FTF survey. Four questions helped establish the eligibility of the respondent, followed by 5–6 demographic questions, 5 questions assessing awareness of contraceptive methods, and 3–4 questions on contraceptive use. The RDD questions were identical to the FTF questions with a few adaptations for phone administration. The RDD survey was available in French and four local languages: Dioula, Fulfulde, Gourmantchema and Moore. The questionnaire is available as a supporting file.

## Measures—Defining call outcomes for RDD survey using AAPOR final disposition codes

We used the 9[th] edition American Association for Political Opinion Research (AAPOR) final disposition case codes to classify call outcomes. [24] The invalid phone numbers identified during the screening process were not assigned a disposition code. The 13 disposition codes were divided into four groups (Not Eligible; Unknown Eligibility–non-interview; Eligible–non-interview; Interview) and are detailed in Table 2.

Non-eligible respondents were categorized into four codes. Respondents were ineligible if they were male, or were female but >49 years and <15 years or did not speak one of the five survey languages. The fourth group consisted of women who spoke one of the survey languages and were between the ages of 15–49 but were ineligible due to quota restrictions.

**Table 2. Final disposition codes for RDD survey, among valid phone numbers.**

| AAPOR Code | Title | Definition | N (42,726) | % |
|---|---|---|---|---|
| | **Not Eligible** | | | *(44.9%)* |
| (4.71) | Gender (not female) | Male | 15,570 | 36.4 |
| (4.72) | Age | Female and age <15 or >49 years | 479 | 1.1 |
| (4.73) | Language | Female and none of the 7 languages available in survey | 326 | 0.8 |
| (4.8) | Quota Filled | Respondent was female and age-eligible but due to quota restrictions was not interviewed | 2,812 | 6.6 |
| | **Unknown Eligibility, Non-Interview** | | | *(49.3%)* |
| UH (3.13) | No Answer | Phone call not picked-up | 18,182 | 42.5 |
| UH (3.14) | Telephone answering device | Phone call went to voice mail | 370 | 0.9 |
| UH (3.21) | No screener completed–talked with respondent but hung-up or refused | Respondent picked- up the call but interviewer was unable to confirm eligibility | 1,984 | 4.6 |
| UO (3.90) | Other (Language not matched with interviewer) | Respondent spoke one of seven survey languages but the interviewer did not speak the same language | 549 | 1.3 |
| | **Eligible, Non-Interview** | | | *(0.17%)* |
| R (2.111) | Refusal pre-consent but confirmed female and 15–49 | Eligible respondent refused to participate before consent | 37 | 0.09 |
| R (2.11) | Refusal at consent | The respondent refused the study during consent | 6 | 0.01 |
| R (2.1) | Break-off (consented but less than 50% of relevant questions answered) | The respondent consented but answered less than 50% of the questions | 32 | 0.07 |
| | **Interview** | | | *(5.6%)* |
| P (1.2) | Partial (50–80% of relevant questions answered) | The respondent consented and answered between 50–80% of the questions | 54 | 0.1 |
| I (1.1 | Complete (more than 80% of relevant questions answered) | The respondent consented and answered more than 80% of the survey questions | 2,325 | 5.4 |

Unknown eligibility was captured in four disposition codes including calls that were never picked up that were classified as "No answer" and calls answered by a voice mailbox, classified as "Telephone answering device". Respondents who answered but for whom age, gender or area of residence was not known were classified as "No screener completed". Finally, respondents who spoke one of the five survey languages but who did not speak the same language as the interviewer and were not reached during subsequent attempts were classified as "Other (language not matched with interviewer)".

The next group "eligible, not interviewed", was divided into three codes and consisted of women ages 15–49 who spoke one of the five survey languages and were not excluded due to quota restrictions. The first was refusal before consent. The second was refusal at consent and the third was "break-off", corresponding to a consenting respondent who completed less than 50% of questions.

The final group included women who completed the interview, classified as a partial interview when 50–80% of questions were answered and a complete interview when 80% of questions or more were answered.

Based on the aforementioned call disposition codes, we created four key call outcome indicators aligned with AAPOR standards–response rate, cooperation rate, refusal rate and contact rate (Table 3). To improve on the specificity of these outcome measures, AAPOR also recommends calculating rates that exclude an estimated number of unknown eligibility phone numbers from the denominator for response and contact rates. Based on pilot data collected in February 2018, we estimated that 20% of calls with unknown eligibility would in fact include an eligible woman. Applying this correction, we defined corrected response rates (Response rates 3 & 4) and a corrected contact rate (Contact rate 2) excluding 80% of attempted calls with unknown eligibility from the denominators.

**Table 3. Call outcome rates for RDD survey based on final disposition distributions.**

| Response Rates | Explanation | Result |
|---|---|---|
| Response rate 1: $\frac{I}{I+P+R+NC+O+UH+UO}$ | *Minimum response rate.* All individuals who complete more than 80% of survey / All attempted calls | 9.9% |
| Response rate 2: $\frac{I+P}{I+P+R+NC+O+UH+UO}$ | All individuals who complete more than 50% of survey / All attempted calls | 10.1% |
| Response rate 3: $\frac{I}{I+P+R+NC+O+0.2*(UH+UO)}$ | All individuals who complete more than 80% of survey / All attempted calls minus 80% of calls with unknown eligibility | 68.0% |
| Response rate 4: $\frac{I+P}{I+P+R+NC+O+0.2*(UH+UO)}$ | All individuals who complete more than 50% of survey / All attempted calls minus 80% of calls with unknown eligibility | 70.4% |
| **Cooperation Rates** | | |
| Cooperation rate 1: $\frac{I}{I+P+R+O}$ | All individuals who complete more than 80% of survey / Eligible individuals who were ever contacted | 94.7% |
| Cooperation rate 2: $\frac{I+P}{I+P+R+O}$ | All individuals who complete more than 50% of survey / Eligible individuals who were ever contacted | 97.1% |
| **Refusal Rate** | | |
| Refusal rate 3: $\frac{R}{I+P+R+NC+O}$ | All individuals who refused to complete the survey / All attempted calls | 3.1% |
| **Contact Rate** | | |
| Contact rate 1: $\frac{I+P+R+O}{I+P+R+NC+O+UH+UO}$ | All phone numbers that answered the call / All phone numbers | 10.4% |
| Contact rate 2: $\frac{I+P+R+O}{I+P+R+NC+O+0.2*(UH+UO)}$ | All phone numbers that answered the call / All attempted calls minus 80% of calls with unknown eligibility | 74.9% |

## Measures—Independent & dependent variables

The outcome of interest was a binary measure of modern contraceptive use, based on two questions that were asked identically in the two surveys. The first question asked whether the respondent or her partner was currently using a form of contraception ("Are you or your partner currently doing something or using any method to delay or avoid getting pregnant?"). If the respondent responded affirmatively, she was asked to specify the type of method used ("Which method or methods are you using?"). If the respondent identified a modern method (as specified below), she was classified as a user of modern contraception.

Traditionally, measures of modern contraceptive use include all modern contraceptives available in a country. In Burkina Faso, 12 modern contraceptive methods are available: male and female sterilization, implant, Intra Uterine Device (IUD), injectables, pill, emergency contraception, male condom, female condom, diaphragm, foam/jelly and the lactation amenorrhea method. However, the RDD survey only collected data on the five most common methods used in Burkina Faso, based on PMA2020 R5 estimates (covering 98.8% of modern method use): implants, injectables, pills, condoms and IUDs. Thus we limited the definition of modern contraceptive use to these five methods for both the FTF and RDD surveys in this study [18]. In addition, we constructed a five-category indicator of method mix, including the following contraceptive methods: implant, IUD, injectables, pills, and condoms.

We selected independent measures related to modern contraceptive use as cited in the literature as well as factors related to phone ownership, that were collected in both surveys to conduct our analysis [25–27]. The independent variable of interest was mode of data collection. The FTF survey was the reference group; RDD the comparison group. Women's sociodemographic characteristics included age grouped in 5-year increments, current union status (in union–i.e., currently married or living with a partner vs. not in union), parity (ever had a birth yes-no). residential area (urban vs. rural), educational attainment (none, primary, or secondary and higher) and language of survey (Moore, French, Dioula, Fulfulde, or Gourmantchema).

**Missing data.** The RDD data had item non-response due to internet outages at the call center (electricity brown-outs).We used the hot deck method [28] to impute missing values for three variables: age (43 missing values, 1.8%), residence (10 missing values, 0.4%) and education (10 missing values, 0.4%), assuming data were missing completely at random.

## Analysis

We first describe RDD call outcomes to evaluate response rate, cooperation rate, refusal rate and contact rate. To examine distributions of the aforementioned independent variables, we conducted univariate analysis, looking at patterns of response in the FTF and RDD samples. Comparisons were made using RDD respondents who completed at least 50% of the questionnaire (N = 2,379) and all women in the FTF sample who represent the target population (n = 3,659), as well as a subsample of FTF respondents who own a cell phone who represent the sample frame of the RDD survey phone (n = 2,027). The R5 data were adjusted for sampling weights, which address disproportionate two-stage cluster sampling and non-response rates. [29] To account for sample distortion based on age, area of residence and level of education, we created post-stratification weights for the RDD sample, using R5 as the reference population.

We examined the equivalence of modern contraceptive use prevalence estimates in the weighted RDD and FTF samples, setting an equivalence margin $\overline{\delta}$ to +/- 4%. The null hypothesis assumed a difference of more than 4% between the two survey estimates. We report the

90% confidence interval for the difference in point estimates, which simulates performing two one-sided tests. We also report a p-value from an adjusted Wald test [30].

We further compared modern contraceptive use by mode of data collection by conducting multivariable logistic regression adjusting for additional sociodemographic factors. We first assessed bi-variate relationships between each co-variate and modern contraceptive use among the pooled FTF-RDD data. We then conducted multivariable logistic regression to assess the odds of modern contraceptive use by survey mode, adjusting for covariates. We also compared the RDD and FTF phone owner sample using multivariable logistic regression. Analysis was performed using weighted RDD and FTF data. We conducted analyses in Stata version 15 (StataCorp 2017) and determined statistical significance using an alpha of less than 0.05.

## Results

### Call outcomes

Approximately 202,295 unique phone numbers were screened of which 21% were deemed valid. The 42,726 valid phone numbers were called by interviewers over the course of a month and constitute our sample size for all RDD survey response outcome analyses. Overall, 45% of the 42,726 valid phone calls were categorized as ineligible, mostly due to a man answering the call (36%) and approximately 6% of calls were non-eligible because of quota restrictions (Table 2). Another 49% of calls fell in the "unknown eligibility" category, the majority of which were call no answer (43%). Less than 1% of eligible women did not complete the survey.

Altogether, 2,379 women completed 50% or more of the survey questions and comprised our RDD study population for contraceptive prevalence analysis (n = 54 were partial completers).

### Survey outcome rates

The minimum response rate (Response rate 1), which includes only interviews where more than 80% of questions were answered and includes all attempted calls in the denominator was 9.9% (Table 3). This percentage rose to 10.1% when including partial survey completion (50–80%). When excluding 80% of unknown eligibility calls from the denominator, response rates increased substantially, to 68% when counting complete interviews in the numerator (Response rate 3) and 70% when also counting partial interviews (Response rate 4).

The cooperation rate, which includes only eligible calls in the denominator, was 94.7% when counting complete interviews (Cooperation rate 1), and 97.1% when also counting partial interviews (Cooperation rate 2). The refusal rate was 3.1%.

### Comparison of FTF & RDD samples

Among the 3,659 women who completed PMA2020 R5 FTF survey, 78% were rural, 72% were married and 75% had given birth (Table 4). The average age of the respondents was 28.6 years (standard error = 0.24). Close to two-thirds of women (63%) had never been to school; thus only one in five had a secondary education or higher. Interviews were most often conducted in Moore (44%) while 10% of women completed the survey in French and 20% completed the interview in another language. 46% of women (N = 1,671) in the FTF survey indicated that they owned a phone. A greater percent of female phone owners lived in an urban area (35%) than the full FTF sample (20%). 30% of FTF phone owners had secondary education or higher and more than half completed the survey in Moore (53%) while 19% completed the survey in French.

**Table 4. Characteristics of women by survey mode and cell phone ownership.**

| Variable | FTF All Respondents | | FTF Phone Owners | | RDD Unweighted | | RDD Weighted* | |
|---|---|---|---|---|---|---|---|---|
| | N = 3,659 | % | N = 1,671 | % | N = 2,379 | % | N = 2,379 | % |
| **Age** | | | | | | | | |
| Mean (standard error) | 28.6 (0.24) | | 28.9 (0.26) | | 27.5 (0.18) | | 28.7 (0.21) | |
| 15–19 | 820 | 22.4 | 316 | 18.9 | 519 | 22.0 | 502 | 21.3 |
| 20–24 | 622 | 17.0 | 311 | 18.6 | 476 | 20.2 | 389 | 16.5 |
| 25–29 | 622 | 17.0 | 316 | 18.9 | 419 | 17.8 | 387 | 16.4 |
| 30–34 | 527 | 14.4 | 246 | 14.7 | 375 | 15.9 | 347 | 14.7 |
| 35–39 | 428 | 11.7 | 204 | 12.2 | 229 | 9.7 | 288 | 12.2 |
| 40–44 | 388 | 10.6 | 172 | 10.7 | 203 | 8.6 | 269 | 11.4 |
| 45–49 | 252 | 6.9 | 100 | 6 | 138 | 5.9 | 177 | 7.5 |
| **Residential area** | | | | | | | | |
| Rural | 2,869 | 78.4 | 1079 | 64.6 | 1,776 | 75.3 | 1,769 | 75.0 |
| Urban | 790 | 21.6 | 592 | 35.4 | 583 | 24.7 | 590 | 25.0 |
| **Marital status** | | | | | | | | |
| Currently not in union | 1,025 | 28.0 | 521 | 31.5 | 601 | 25.6 | 514 | 21.8 |
| Currently in union | 2,634 | 72.0 | 1145 | 68.5 | 1,748 | 74.4 | 1,845 | 78.2 |
| **Highest school attended** | | | | | | | | |
| Never | 2,334 | 63.8 | 869 | 52 | 1,210 | 51.3 | 1,484 | 62.9 |
| Primary | 593 | 16.2 | 296 | 17.7 | 387 | 16.4 | 392 | 16.6 |
| Secondary or higher | 732 | 20.0 | 508 | 30.3 | 762 | 32.3 | 484 | 20.5 |
| **Parity** | | | | | | | | |
| Avg # of kids among parous women | 3.0 | | 2.7 | | 3.2 | | 3.6 | |
| No | 918 | 25.1 | 434 | 26 | 573 | 24.4 | 488 | 20.7 |
| Yes | 2,741 | 74.9 | 1237 | 74 | 1,774 | 75.6 | 1,871 | 79.3 |
| **Language** | | | | | | | | |
| Dioula | 388 | 10.2 | 373 | 11.1 | 169 | 7.2 | 193 | 8.2 |
| French | 1,603 | 10.6 | 388 | 18.5 | 600 | 25.6 | 455 | 19.3 |
| Fulfulde | 377 | 4.9 | 179 | 0.67 | 27 | 1.15 | 31 | 1.3 |
| Gourmantchema | 179 | 10.3 | 377 | 6.2 | 25 | 1.1 | 28 | 1.2 |
| Moore | 373 | 43.8 | 1602 | 53.3 | 1,521 | 64.9 | 1,651 | 70.0 |
| Other | 743 | 20.3 | 169 | 10.1 | - | - | - | - |
| **Province** | | | | | | | | |
| Boucle du Mouhoun | 410 | 11.2 | 130 | 7.8 | 134 | 5.8 | 134 | 5.7 |
| Cascades | 168 | 4.6 | 42 | 2.5 | 73 | 3.2 | 73 | 3.1 |
| Centre | 395 | 10.8 | 326 | 19.5 | 574 | 25.0 | 578 | 24.5 |
| Centre-Est | 267 | 7.3 | 125 | 7.5 | 206 | 9.0 | 217 | 9.2 |
| Centre-Nord | 351 | 9.6 | 155 | 9.3 | 240 | 10.4 | 255 | 10.8 |
| Centre-Ouest | 417 | 11.4 | 207 | 12.4 | 170 | 7.4 | 172 | 7.3 |
| Centre- Sud | 99 | 2.7 | 38 | 2.3 | 99 | 4.3 | 104 | 4.4 |
| Est | 424 | 11.6 | 137 | 8.2 | 131 | 5.7 | 130 | 5.5 |
| Hauts-Bassins | 333 | 9.1 | 204 | 12.2 | 206 | 9.0 | 210 | 8.9 |
| Nord | 304 | 8.3 | 170 | 10.2 | 177 | 7.7 | 189 | 8.0 |
| Plateau-Central | 124 | 3.4 | 75 | 4.5 | 158 | 6.9 | 170 | 7.2 |
| Sahel | 238 | 6.5 | 30 | 1.8 | 78 | 3.4 | 78 | 3.3 |
| Sud-Ouest | 1 | 0.04 | 30 | 1.8 | 33 | 2.31 | 50 | 2.1 |

FTF analyses are weighted for survey design-weight.

*post-stratification weights included age, residence and level of education

Compared to the target population of FTF respondents, women in the RDD sample were younger, a greater percent had a secondary degree and responded in French. Distribution by region also differed with greater representation of women in the Center region (where the capital, Ouagadougou, is located) in the RDD sample. Characteristics of women in the RDD sample more closely reflected the sample FTF phone owners. After applying post-stratification weights, age and educational differences disappeared but regional differences increased.

## Contraceptive use by survey mode

**Contraceptive use among FTF, phone owner and RDD weighted samples.** A quarter of women (26% (95% Confidence Interval (CI): 22.7%– 29.6%)) in the FTF survey reported contraceptive use versus 40.2% (95% CI: 38.2%– 42.4%) in the unweighted RDD sample. The post-stratification weights had little impact on the RDD estimate: 38.7% contraceptive use (95% CI: 36.7%– 40.8%). The 12.7% difference was greater than the 4% equivalence margin leading to conclude RDD and FTF estimates are not equivalent. The difference between RDD and FTF phone owners was smaller but remained substantial (40.2% to 31.7%).

Method mix differed by mode of data collection. Altogether 46% of contraceptive users in the FTF sample used a short-acting method (injectables, pills, or condoms) and 54% used long acting reversible contraception (LARC), including IUD or implant (Fig 1) compared to 59% and 41% in the RDD sample respectively. The RDD sample estimates more closely resembled the phone FTF method mix of 55% short-acting method and 45% LARC.

After adjusting for confounding covariates, results from the multivariable logistic regression model indicated that women in the weighted RDD sample had almost twice the odds of

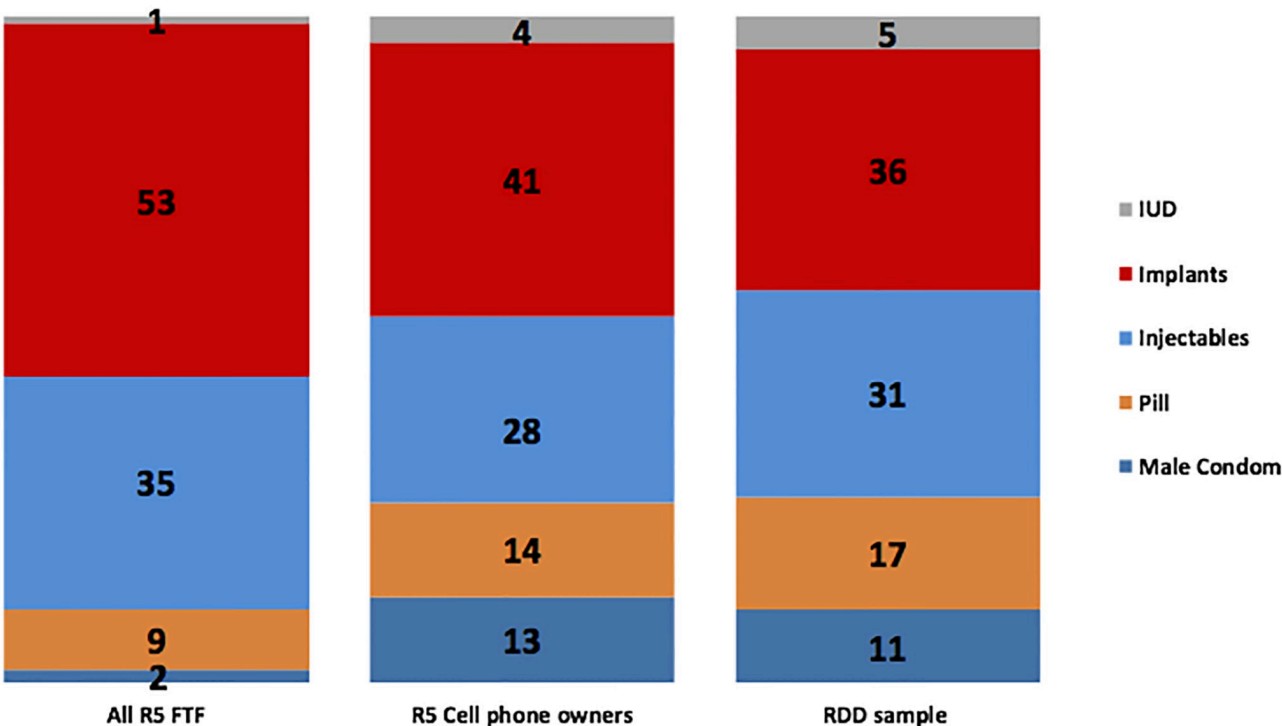

**Fig 1. Modern method mix among current users among full FTF sample, FTF cell phone owners, and RDD respondents* (%).** *The most effective method currently used, if multiple methods were reported. % estimates are adjusted for sampling weight.

reporting modern contraceptive use compared to women in the FTF sample (Odds Ratio (OR): 1.9, 95% CI: 1.6–2.1) (Table 5). Significant differences also remained when comparing RDD and FTF phone owners (OR: 1.6 95% CI: 1.3–1.8).

## Discussion

This study assessed the feasibility of conducting RDD phone surveys for monitoring family planning metrics in Burkina Faso and the associated bias in estimating national modern contraceptive prevalence. We found that six percent of valid phone numbers resulted in a complete interview. The RDD sample resulted in 14% point over-estimation of modern contraceptive use, which remained substantial and significant, after post-stratification adjustments of the

**Table 5. Odds of reporting modern contraceptive use by women's characteristics and survey mode, using full FTF sample and FTF phone owner sample.**

|  | RDD vs. Full FTF sample | | RDD vs. FTF phone owner sample |
|---|---|---|---|
|  | **Unadjusted OR (95% CI)** | **Adjusted OR (95% CI)** | **Adjusted OR (95% CI)** |
| **Mode** |  |  |  |
| FTF *(reference)* |  |  |  |
| RDD | **1.8 (1.6–2.0)** | **1.9 (1.6–2.2)** | **1.6 (1.3–1.8)** |
| **Age group** |  |  |  |
| 15–19 *(reference)* |  |  |  |
| 20–24 | **2.4 (2.0–3.0)** | **1.4 (1.1–1.8)** | **1.4 (1.1–1.8)** |
| 25–29 | **3.0 (2.5–3.8)** | **1.6 (1.2–2.1)** | **1.6 (1.3–2.1)** |
| 30–35 | **3.4 (2.7–4.2)** | **1.7 (1.3–2.3)** | **1.6 (1.2–2.1)** |
| 35–39 | **3.3 (2.6–4.2)** | **1.6 (1.2–2.2)** | **1.7 (1.2–2.3)** |
| 40–44 | **2.2 (1.7–2.8)** | 1.1 (0.8–1.5) | 1.1 (0.8–1.6) |
| 45–49 | 1.0 (0.7–1.4) | 0.5 (0.4–0.7) | 0.5 (0.4–0.8) |
| **Residential area** |  |  |  |
| Rural *(reference)* |  |  |  |
| Urban | **1.3 (1.1–1.4)** | 1.0 (0.9–1.2) | 0.9 (0.8–1.2) |
| **Highest school attended** |  |  |  |
| No education *(reference)* |  |  |  |
| Primary | **1.5 (1.3–1.8)** | **1.7 (1.5–2.1)** | **1.7 (1.4–2.1)** |
| Secondary or more | **1.3 (1.1–1.5)** | **1.9 (1.5–2.3)** | **1.6 (1.3–2.0)** |
| **Survey Language** |  |  |  |
| French *(reference)* |  |  |  |
| Moore | **0.6 (0.6–0.7)** | **0.6 (0.5–0.8)** | **0.6 (0.5–0.8)** |
| Gourma | **0.7 (0.5–0.9)** | 0.9 (0.7–1.3) | 1.0 (0.7–1.6) |
| Fulfulde | **0.2 (0.1–0.3)** | **0.3 (0.2–0.5)** | 0.5 (0.3–1.3) |
| Dioula | 0.8 (0.6–1.0) | 0.8 (0.7–1.1) | 0.9 (0.7–1.3) |
| Other | **0.6 (0.5–0.8)** | 0.9 (0.7–1.3) | 0.9 (0.6–1.5) |
| **Ever Birth** |  |  |  |
| Never given birth *(reference)* |  |  |  |
| Ever given birth | **3.7 (3.2–4.4)** | **4.8 (3.8–6.0)** | **3.7 (3.0–4.6)** |
| **Ever married** |  |  |  |
| Not married (reference) |  |  |  |
| Married | **1.7 (1.5–2.0)** | 0.9 (0.7–1.0) | **0.8 (0.6–0.9)** |

Bold denotes a p-value < = 0.05

RDD sample and further adjustment for confounding (OR: 1.9, 95% CI: 1.6–2.2). Whereas previous RDD studies in LMIC have compared their phone-based estimates with non-concurrent studies that were managed by exterior survey organizations, our study used the same questionnaire for both the FTF and RDD study, within a reasonable time frame, allowing us to more confidently compare the outcomes. This study is among the first in SSA to offer this direct comparison of modes and to look specifically at women.

As expected from previous studies on phone ownership [7, 31] and from our previous work comparing respondents to non-respondents among phone owners [19], the RDD sample was distorted in favor of more educated women. In addition, the RDD sample over-represented women speaking Moore and living in Ouagadougou. The application of quotas, which excluded only six percent of eligible respondents, helped reduce age and residence distortions.

We nonetheless applied post-stratification weights to address distortion according to educational, age and urban/rural residence. The weights had little impact on the difference in contraceptive prevalence rates between RDD and FTF (14.2% point difference using unweighted RDD estimates versus 12.7% point difference using weighted RDD estimates). A trio of articles from Brazil, all using data from VIGITEL, an annual and continuous phone survey in the 26 state capitals monitoring a host of non-communicable diseases, compared RDD samples with concurrent FTF samples [32–34]. Two of the articles used post-stratification weights and reported the weights reduced the difference between phone survey estimates and FTF survey estimates [32, 33].

To better understand the reasons for the RDD over-estimation of modern contraceptive use in our study, we assessed the impact of frame bias by comparing RDD estimates to FTF phone owner estimates. The difference between the RDD unweighted sample and the FTF subsample of phone owners was significantly reduced (8.5% point difference) compared to the 14.2% difference observed with the full FTF sample, suggesting frame bias may contribute for more than half of the over-estimation of modern contraceptive prevalence in the RDD sample. These results are in line with previous analyses of female phone owners in Burkina Faso, showing greater use of contraception among cell phone owners compared to non-owners [26]. They are also consistent with the findings of one of the aforementioned Brazilian studies showing little difference between RDD estimates and FTF estimates among phone owners [32].

Nonetheless, the difference in estimates of modern contraceptive use between FTF phone owners and RDD respondents (unweighted) suggests additional sources of bias. The bias is likely caused by differences in population composition (non-response) or differences in survey response by mode of data collection (measurement error). The RDD sample showed greater representation of women with secondary education or higher, women under 34 and French- and Moore-speaking women compared to the FTF sample. However, after adjusting for these factors in the multivariable analysis, the odds of contraceptive use remained significantly higher in the unweighted RDD sample compared to the FTF phone owner sample (OR: 1.6 95% CI: 1.3–1.8). Unfortunately, we were not able to collect community-level variables that are frequently associated with contraceptive use, such as quality of and access to health services, cultural norms and fertility practices.

This study has a number of strengths. It is among the first in SSA to compare health estimates from concurrent surveys using different modes of data collection. As such, it provides a firsthand investigation of the opportunities and challenges of using phone surveys in a context of rapid demographic change. The use of similar questionnaires limited measurement error while the almost concurrent timing of the surveys also improved comparability of survey estimates. The sample sizes were large enough to allow equivalence testing of modern contraceptive use prevalence with a relatively low margin of equivalence of 4%. Furthermore, this study is among the first in LMIC to use CATI during RDD; traditionally IVR is used with RDD

sampling in LMIC. The results, which show that cell phone surveys among women in West Africa do not create estimates comparable to a concurrent FTF survey, provide a warning to those who are eager to replace FTF surveys with cell phone surveys.

The study also has a number of limitations, including the small number of demographic variables available in the RDD sample, which limited our capacity to systematically investigate differences between the RDD sample and the FTF target population. As a result, post-stratification weights were limited to a few demographics, leaving out potential unobserved differences that could better explain the difference between RDD and FTF modern contraceptive use estimates. Although we used multiple weights to correct for the difference, the modern contraceptive use estimate from the RDD survey remained higher than the estimate from the FTF survey. The study could not explain all the factors that may cause the differences in estimates between two surveys. Another limitation was not allowing men to pass the phone to a female respondent as 84% of households own a phone but fewer females own their own phone. Although the decision to not pass the phone was made to reduce the complexity of weighting (i.e. avoiding weighting the sample for women in a household that were not sampled), the volume of calls we had to place due to men answering the majority of calls (36.4% (N = 15,570) of calls picked up by men) made the project very challenging to implement and may have impacted data quality.

## Conclusion

An RDD survey in Burkina Faso did not yield an estimate of modern contraceptive use that was equivalent to FTF reference estimate, even after applying post-stratification weights. Over-estimation of modern contraceptive use in our RDD survey originated both from a truncated sample frame, excluding non-cell phone owners, and also from non-response and measurement error, which need further examination as cell phone ownership expands in the SSA.

## Supporting information

**S1 Data.**
(DOCX)

## Acknowledgments

The authors would like to acknowledge three groups of people that made this research possible: first, the female survey respondents, who were willing to participate in a new form of data collection. Second, the interviewers and call center supervisors who showed great perseverance and dedication during a challenging data collection. Finally, we would like to show our appreciation for Viamo, the organization that created the software for data collection.

## Author Contributions

**Conceptualization:** Aliou Gadiaga, Georges Guiella, Shani Turke, Noelle Battle, Saifuddin Ahmed, Caroline Moreau.

**Data curation:** Abigail R. Greenleaf.

**Formal analysis:** Abigail R. Greenleaf, Saifuddin Ahmed, Caroline Moreau.

**Funding acquisition:** Georges Guiella, Saifuddin Ahmed, Caroline Moreau.

**Investigation:** Abigail R. Greenleaf, Aliou Gadiaga, Shani Turke, Noelle Battle.

**Methodology:** Abigail R. Greenleaf, Caroline Moreau.

**Project administration:** Aliou Gadiaga, Georges Guiella.

**Resources:** Abigail R. Greenleaf, Aliou Gadiaga.

**Software:** Abigail R. Greenleaf.

**Supervision:** Abigail R. Greenleaf, Saifuddin Ahmed, Caroline Moreau.

**Validation:** Abigail R. Greenleaf, Saifuddin Ahmed.

**Visualization:** Abigail R. Greenleaf.

**Writing – original draft:** Abigail R. Greenleaf, Saifuddin Ahmed, Caroline Moreau.

**Writing – review & editing:** Shani Turke, Noelle Battle.

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
