## [Decision Letter · Decision Letter 0]

28 Nov 2019

PONE-D-19-19653

Comparability of modern contraceptive use estimates between a face-to-face survey and a cellphone survey among women in Burkina Faso

PLOS ONE

Dear Mr. Greenleaf,

Thank you for submitting your manuscript to PLOS ONE. After careful consideration, we have decided that your manuscript does not meet our criteria for publication and must therefore be rejected.

Specifically: there is lack of consistent information, at some places the statements are not clearly presented. The findings and the interpretation of data dont match and variables used in the research are not clearly defined.

I am sorry that we cannot be more positive on this occasion, but hope that you appreciate the reasons for this decision.

Yours sincerely,

Russell Kabir, PhD

Academic Editor

PLOS ONE

Reviewers' comments:

Reviewer's Responses to Questions

**Comments to the Author**

1. Is the manuscript technically sound, and do the data support the conclusions?

Reviewer #1: Yes

Reviewer #2: Yes

Reviewer #3: Partly

2. Has the statistical analysis been performed appropriately and rigorously? 

Reviewer #1: Yes

Reviewer #2: Yes

Reviewer #3: No

3. Have the authors made all data underlying the findings in their manuscript fully available?

Reviewer #1: Yes

Reviewer #2: Yes

Reviewer #3: Yes

4. Is the manuscript presented in an intelligible fashion and written in standard English?

Reviewer #1: Yes

Reviewer #2: Yes

Reviewer #3: No

5. Review Comments to the Author

Reviewer #1: There are few comments as well such as:

1- What are the information in first column? Are they age categories?

2- Page 15, line 317, “significance using an alpha of 0.05” should be written “less than 0.05”.

3- Page 20, line 347, please include SD or SE when you report the mean.

4- Table 4, please clarify different marital status as married and non-married.

5- Table 5 can be deleted as its information has been explained in text completely.

6- Table 6, some figured are bold which probably shows the significance but p-value is not reported.

7- Discussion should show the application of the study results.

Reviewer #2: The authors present an important issue relevant for public health and epidemiological data collection. In the present article, the main aim was to assess survey errors when comparing computer assisted telephone interviews (CATI) to FTF surveys for modern contraceptive use in Burkina Faso. As the first of its kind, this paper is essential given the recent increase in cell phone ownership in low to middle income countries (LMIC). I commend their brilliant presentation and recommend this article to be considered for publication.

My only comment relating to this article is given below:

The authors clearly pointed out that the difference in contraceptive prevalence rates could not be fully explained owing in part to the small number of demographic variables available in the RDD survey. Other key indicators that influence the use of modern contraceptives in LMIC settings include the health service environment, physical infrastructure, and prevailing cultural beliefs surrounding health care seeking. Were any of these assessed as part of the study? It will be interesting to see how any of these other indicators contribute to the difference in prevalence rates between the two surveys.

Reviewer #3: I do not think the manuscript is written in a scientific manner and there is a lack of consistency across the manuscript. The comparison of both data and their interpretations are difficult to understand. Moreover, this is not a suitable manuscript for Plos One readers.

6. PLOS authors have the option to publish the peer review history of their article (what does this mean?). If published, this will include your full peer review and any attached files.

Reviewer #1: No

Reviewer #2: No

Reviewer #3: No

- - - - -

---

## [Author Response · Author response to Decision Letter 0]

18 Jan 2020

Please find response to Reviewers 1-3, and the Academic Editor. I am submitting after receiving approval for an appeal and re-submission. 

Reviewer #1: 

1- What are the information in first column? Are they age categories?

Thank you for your careful attention to Table 1. Yes, this column is age groups. I have changed the column heading accordingly. Line 159 

2- Page 15, line 317, “significance using an alpha of 0.05” should be written “less than 0.05”.

We have made this change. Line 310 

3- Page 20, line 347, please include SD or SE when you report the mean.

We added the standard error to the text that refers to average age. Line 341 

4- Table 4, please clarify different marital status as married and non-married.

We prefer to refer to our respondents as “in union” or “not in union” as in the setting (Burkina Faso) many people may not be married but are co-habiting, which we refer to as “in union” because they are not legally married. We will keep this classification as is, as this is also the classification the Demographic and Health Survey uses, which is our reference standard. Lines 276, 277 and Table 4. 

5- Table 5 can be deleted as its information has been explained in text completely.

Point well taken. I have deleted Table 5. 

6- Table 6, some figured are bold which probably shows the significance but p-value is not reported.

We have added a footnote at the bottom of the table that the bold denotes statistical significance less than 0.05. We report the confidence intervals instead of the p-value as we find the confidence interval to be more telling information than the p-value. Table 6. 

7- Discussion should show the application of the study results.

We added a more explicit explanation of the application of study results in two areas of the discussion section: lines 411 – 414 & lines 464 – 466. 

Reviewer #2: 

The authors present an important issue relevant for public health and epidemiological data collection. In the present article, the main aim was to assess survey errors when comparing computer assisted telephone interviews (CATI) to FTF surveys for modern contraceptive use in Burkina Faso. As the first of its kind, this paper is essential given the recent increase in cell phone ownership in low to middle income countries (LMIC). I commend their brilliant presentation and recommend this article to be considered for publication.

My only comment relating to this article is given below:

The authors clearly pointed out that the difference in contraceptive prevalence rates could not be fully explained owing in part to the small number of demographic variables available in the RDD survey. Other key indicators that influence the use of modern contraceptives in LMIC settings include the health service environment, physical infrastructure, and prevailing cultural beliefs surrounding health care seeking. Were any of these assessed as part of the study? It will be interesting to see how any of these other indicators contribute to the difference in prevalence rates between the two surveys.

Thank you for your feedback. Unfortunately we did not collect data on key indicators you mention above. We added a sentence in the discussion section about how these variables could better explain the difference in our outcomes of interest. See lines 448-450.

Reviewer #3: 

I do not think the manuscript is written in a scientific manner and there is a lack of consistency across the manuscript. The comparison of both data and their interpretations are difficult to understand. Moreover, this is not a suitable manuscript for Plos One readers.

Academic Editor: 

1) There is lack of consistent information

This statement does not provide details about where there is perceived inconsistent information 

2) At some places the statements are not clearly presented

This statement does not provide details about where there is perceived inconsistent information 

3) The findings and the interpretation of data don’t match

This statement does not provide details about where there is perceived inconsistent information 

4) Variables used in the research are not clearly defined

The AAPOR variables used (Subheading “Measure – defining call outcomes from RDD survey using AAPOR final disposition codes”) follow the same guidelines as used by L’Engle et al. published by PLOSOne in 2018. Furthermore, the outcome variables also follow the AAPOR guidelines. 

The dependent variable, modern contraceptive use, follows the definition of modern contraceptive use given by the Demographic and Health Survey, which is the reference standard. The independent variables, which are commonly used socio-demographic variables, are defined between lines 272 – 279.

---

## [Editor Report · Decision Letter 1]

2 Apr 2020

Comparability of modern contraceptive use estimates between a face-to-face survey and a cellphone survey among women in Burkina Faso

PONE-D-19-19653R1

Dear Dr.  Abigail Greenleaf,  

After reviewing your manuscript, your appeal and all reviewer comments, we are pleased to inform you that your manuscript has been judged scientifically suitable for publication and will be formally accepted for publication once it complies with all outstanding technical requirements.

With kind regards,

Kenneth Ngure, PhD MPH

Academic Editor

PLOS ONE

Tanya Doherty PhD

Academic Editor

PLOS ONE

Journal Requirements:

1. Please include additional information regarding the survey or questionnaire used in the RDD study and ensure that you have provided sufficient details that others could replicate the analyses. For instance, if you developed a questionnaire as part of this study and it is not under a copyright more restrictive than CC-BY, please include a copy, in both the original language and English, as Supporting Information.

2. Please note that in order to use the direct billing option the corresponding author must be affiliated with the chosen institute. Please either amend your manuscript or remove this option (via Edit Submission).

4. Thank you for including your ethics statement on the online submission form:

"The PMA2020 FTF survey has exempt status (IRB #00000287, exempt as “public health practice”, defined by DHHS regulations 45 CFR 46.102), as determined by John Hopkins’ IRB. Approval was granted in July 2014 by the Comité d’éthique pour la recherche en santé (IRB #2014-7-81). Consent was obtained orally. For the phone survey, ethical approval was granted by the IRB at the Johns Hopkins Bloomberg School of Public Health (IRB No. 00007961). The ISSP team submitted a protocol to their ethical committee for the RDD survey (IRB No. 2018-3-036).".

* To help ensure that the wording of your manuscript is suitable for publication, would you please also add this statement at the beginning of the Methods section of your manuscript file.

---

## [Editor Report · Acceptance letter]

16 Apr 2020

PONE-D-19-19653R1 

Comparability of modern contraceptive use estimates between a face-to-face survey and a cellphone survey among women in Burkina Faso 

Dear Dr. Greenleaf:

I am pleased to inform you that your manuscript has been deemed suitable for publication in PLOS ONE. Congratulations! Your manuscript is now with our production department. 

With kind regards,

on behalf of

Dr. Kenneth Ngure 

Academic Editor

PLOS ONE